# DIAGNOSING THE ENVIRONMENT BIAS IN VISION-AND-LANGUAGE NAVIGATION

## ABSTRACT

Vision-and-Language Navigation (VLN) requires an agent to follow natural-language instructions, explore the given environments, and reach the desired target locations. These step-by-step navigational instructions are extremely useful in navigating new environments that the agent does not know about previously. Most recent works that study VLN observe a significant performance drop when tested on unseen environments (i.e., environments not used in training), indicating that the neural agent models are highly biased towards training environments. Although this issue is considered as one of the major challenges in VLN research, it is still under-studied and needs a clearer explanation. In this work, we design novel diagnosis experiments via environment re-splitting and feature replacement, looking into possible reasons for this environment bias. We observe that neither the language nor the underlying navigational graph, but the low-level visual appearance conveyed by ResNet features directly affects the agent model and contributes to this environment bias in results. According to this observation, we explore several kinds of semantic representations which contain less low-level visual information, hence the agent learned with these features could be better generalized to unseen testing environments. Without modifying the baseline agent model and its training method, our explored semantic features significantly decrease the performance gap between seen and unseen on multiple datasets (i.e., 8.6% to 0.2% on R2R, 23.9% to 0.1% on R4R, and 3.74 to 0.17 on CVDN) and achieve competitive unseen results to previous state-of-the-art models.

## 1 INTRODUCTION

Vision-and-Language Navigation (VLN) tests an agent's ability to follow complex natural language instructions as well as explore the given environments, so as to be able to reach the desired target locations. As shown in Fig. 1, the agent is put in an environment and given a detailed step-by-step navigational instruction. With these inputs, the agent needs to navigate the environment and find the correct path to the target location. In this work, we focus on the instruction-guided navigation (MacMahon et al., 2006; Anderson et al., 2018b; Misra et al., 2018; Blukis et al., 2018; Chen et al., 2019c) where detailed step-by-step navigational instructions are used (e.g., 'Go outside the dining room and turn left ...'), in contrast to the target-oriented navigation (Gordon et al., 2018; Das et al., 2018; Mirowski et al., 2018; Yu et al., 2019) where only the target is referred (e.g., 'Go to the kitchen' or 'Tell me the color of the bedroom'). Although these step-by-step instructions are over-detailed when navigating local areas (e.g., your home), they are actively used in unseen environments (e.g., your friend's house, a new city) where the desired target is usually unknown to navigational agents. For this purpose, testing on unseen environments which are not used during agent-training is important and widely accepted by instruction-guided navigation datasets.

Recent works propose different methods to improve generalizability of agents on these unseen testing environments; and most of the existing works (Anderson et al., 2018b; Wang et al., 2018b; Fried et al., 2018; Wang et al., 2019b; Ma et al., 2019a;b; Tan et al., 2019; Huang et al., 2019; Hu et al., 2019) observe a significant performance drop from seen environments (i.e., the environments used in training) to unseen environments (i.e., the environments not used in training), which indicates a strong bias in the model towards the training environments. While this performance gap is emphasized as one of the major challenges in current VLN research, the issue is still left unresolved and waits for an explicit explanation. Thus, in this paper, we aim to answer three questions to this

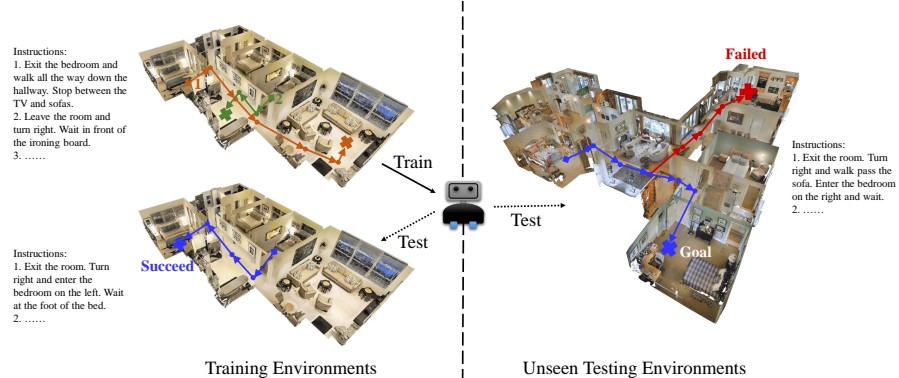

Figure 1: Vision-language-navigation: performance of the agent drops in unseen environments.

environment bias: 1. **Where** (i.e., in which component) is the bias located? 2. **Why** does this bias exist? 3. **How** to eliminate this bias?

To locate **where** the bias is, we start by showing that natural-language navigational instructions and underlying navigational graphs are not direct reasons for this performance gap. We then investigate the effect of environments on the agent's performance. In order to conduct a detailed analysis, we re-split the environment and categorize the validation data into three sets based on their visibility to the training set: *path-seen* data intersecting with the training paths, *path-unseen* data using the training environments but away from the training paths, and *env-unseen* data using unseen environments (environments not used in training). By showing that the results gradually decrease from path-seen data to env-unseen data, we characterize the environment bias at three levels: path level, region level, and environment level.

These three levels of environment biases indicate strong 'spatial localities' in the tasks of VLN, which are intuitively reasonable because environments and regions (e.g., houses and cities) usually have their own styles when built or decorated. We next want to analyze the detailed reason **why** this locality would further lead to a gap in seen versus unseen results. Our hypothesis is that the *low-level information* carried by the ResNet features (He et al., 2016) is the reason. To keep minimal low-level visual information and promote more high-level semantic information, we replace the ResNet features with the 1000 ImageNet classification probabilities. Although the semantic information encoded by these features is not accurate because of the shifted domain of images and labels, the same model with ImageNet-Labels features performs surprisingly well on various VLN datasets (i.e., Room-to-Room, R4R, and CVDN[1]). Most importantly, these noisy semantic features effectively eliminate the performance gap between seen and unseen environments, which suggests that the environment bias is attributed to the ResNet features as our hypothesis.

Following the practice in using ImageNet labels as semantic features, we further provide a discussion on **how** the environment bias could be eliminated. For this, we employ advanced high-level semantic features which are more rational for the VLN domain. We explore three kinds of semantic features: (1) areas of detected object labels (Ren et al., 2015); (2) ground truth semantic views (Chang et al., 2017); and (3) learned semantic view features. We show that all of these semantic features significantly reduce the environment bias in multiple datasets and also achieve strong results in testing unseen environments. We hope this work encourages more investigation and research into improving the generalization of vision-language models to unseen real-world scenarios.

## 2 RELATED WORK

**Vision-and-Language Navigation**: Vision-and-language navigation is an emerging task in the vision-and-language area. A lot of datasets have been proposed in recent years, such as Room-to-Room (Anderson et al., 2018b), Room-for-Room (Jain et al., 2019), TouchDown (Chen et al., 2019c), CVDN (Thomason et al., 2019b), RERERE (Qi et al., 2019), House3D (Wu et al., 2018) and EQA (Das et al., 2018). Recent works (Thomason et al., 2019a; Wang et al., 2018b; Fried et al.,

---

[1]We did not test these semantic features on touchdown (Chen et al., 2019c) since the images are not released.

2018; Wang et al., 2019b; Ma et al., 2019a;b; Tan et al., 2019; Hu et al., 2019; Ke et al., 2019; Anderson et al., 2019) focusing on improving the performance of navigation models, especially in unseen testing environments, have helped to increase the navigational success rate.

**Domain Adaptation**: The general setup of domain adaption contains two sets of data samples $\{x_i\}_{x_i \in X}$ and $\{y_i\}_{y_i \in Y}$ from two domains $X$ and $Y$. Based on these samples, we could learn domain invariant feature with adversarial training (Goodfellow et al., 2014; Zhu et al., 2017; Long et al., 2018; Wang et al., 2019a; Hosseini-Asl et al., 2019; Zhang et al., 2019; Gong et al., 2019; Chen et al., 2019b) or learn a transfer function $f : X \rightarrow Y$ (Wang et al., 2018a; Chen et al., 2019a; Rozantsev et al., 2018). However, samples from the target domain may not be available (e.g., the testing environments in navigation should not be used in training) in applications. Thus, we try to give an interpretable explanation to why performance varies in different domains and design a robust feature for it without deliberately considering the target domain. Two methods in VLN, RCM (Wang et al., 2019b) and EnvDrop (Tan et al., 2019), explore the possibility of domain adaptation. Both works take the testing environments in training while RCM also uses testing instructions.

**Domain Generalization**: In domain generalization (Blanchard et al., 2011), the goal is to predict the labels in the previous unseen domain. Similar to the test setting of VLN tasks, the testing data is unrevealed in training. Works have been proposed to learn the common features of the training domain (Muandet et al., 2013; Blanchard et al., 2017; Li et al., 2017; 2018; Carlucci et al., 2019; Deshmukh et al., 2019). In this paper, we focus on the domain generalization problem in VLN task, and try to find the reasons for the failures.

## 3  VISION-AND-LANGUAGE NAVIGATION AND ITS ENVIRONMENT BIAS

We first introduce the task of vision-and-language navigation (VLN) and briefly describe the neural agent models used in our work. We next survey previous works on multiple indoor navigation datasets to show that the environment bias is widely observed in current VLN research. Lastly, we claim that this bias also exists in the outdoor navigation tasks, if the agent is tested on unseen regions.

### 3.1  VISION-AND-LANGUAGE NAVIGATION

**Tasks**: As shown in Fig. 1, the goal of the VLN task is to train an agent to navigate a certain type of environments $\{\mathbf{E}\}$ (e.g., indoor or outdoor environments) given the instruction $\mathbf{I}$. Each environment $\mathbf{E}$ is an independent space, such as a room or a house, and consists of a set of viewpoints. Each viewpoint is represented as a panoramic image and can be decomposed into separate views $\{o\}$ as inputs to the neural agent models. The viewpoints and their connectivity form the navigational graph. In practice, after being placed at a particular viewpoint and given the instruction in the beginning, at each time step, the agent can observe the panoramic image of the viewpoint where it is located, and choose to move along an edge of the graph to the next node (i.e., viewpoint) or stop. This navigational process produces a path (i.e., a list of viewpoints), and the performance of the agent is evaluated by whether it reaches the target location that the instruction indicates in the end.

**Neural Agent Models**: Most instruction-guided navigational agents are built based on attentive encoder-decoder models (Bahdanau et al., 2015). The encoder reads the instructions while the decoder outputs actions based on the encoded instructions and perceived environments. Since the main purpose of this work is to understand the environment bias in vision-and-language navigation, we use a minimal representative neural agent model that achieves comparable results to previous works. Specifically, we adopt the panoramic-view neural agent model in Fried et al. (2018) ('Follower') with modifications from Tan et al. (2019) as our baseline model. We also exclude advanced training techniques (i.e., reinforcement learning and data augmentation) and only train the agent with imitation learning in all our experiments for the same purpose. More details in original papers.

### 3.2  ENVIRONMENT BIAS IN INDOOR NAVIGATION

In order to evaluate the generalizability of agent models, indoor vision-and-language navigation datasets (e.g., those collected from Matterport3D (Chang et al., 2017)) use disjoint sets of environments in training and testing. Most of the datasets provide two validation splits to verify the agent's

Table 1: Results show the performance gap between seen ('Val Seen') and unseen ('Val Unseen') environments in several VLN tasks. Room-to-Room and Room-for-Room are evaluated with 'Success Rate', CVDN is evaluated with 'Goal Progress', Touchdown is evaluated with 'Task Completion'.

| Task | Method | Result | | |
|------|--------|--------|--------|--------|
| | | **Val Seen** | **Val Unseen** | **Abs Gap** $|\Delta|$ |
| Room-to-Room (Anderson et al., 2018b) | R2R (Anderson et al., 2018b) | 38.6 | 21.8 | 16.8 |
| | RPA (Wang et al., 2018b) | 42.9 | 24.6 | 18.3 |
| | S-Follower (Fried et al., 2018) | 66.4 | 35.5 | 30.9 |
| | RCM (Wang et al., 2019b) | 66.7 | 42.8 | 23.9 |
| | SMNA (Ma et al., 2019a) | 67 | 45 | 22 |
| | Regretful (Ma et al., 2019b) | 69 | 50 | 19 |
| | EnvDrop (Tan et al., 2019) | 62.1 | 52.2 | 9.9 |
| | ALTR (Huang et al., 2019) | 55.8 | 46.1 | 9.7 |
| | RN+Obj (Hu et al., 2019) | 59.2 | 39.5 | 19.7 |
| | CG (Anderson et al., 2019) | 31 | 31 | **0** |
| | Our baseline | 56.1 | 47.5 | 8.6 |
| | **Our learned-semantic** | 53.1 | **53.3** | **0.2** |
| Room-for-Room (Jain et al., 2019) | Speaker-Follower | 51.9 | 23.8 | 28.1 |
| | RCM | 55.5 | 28.6 | 26.9 |
| | Our baseline | 54.6 | 30.7 | 23.9 |
| | **Our learned-semantic** | 36.2 | **36.1** | **0.1** |
| CVDN (Thomason et al., 2019b) | NDH | 5.92 | 2.10 | 3.82 |
| | Our baseline | 5.97 | 2.23 | 3.74 |
| | **Our learned-semantic** | 2.60 | **2.43** | **0.17** |
| Touchdown (Chen et al., 2019c) | GA (original split) | 7.9 (dev) | 5.5 (test) | – |
| | RCONCAT (original split) | 9.8 (dev) | 10.7 (test) | – |
| | Our baseline (original split) | 15.0 (dev) | 14.2 (test) | – |
| | Our baseline (seen/unseen split) | 17.5 | 5.3 | 12.2 |

performance in both sets of environments: validation seen, which takes the data from training environments, and validation unseen, whose data is from new environments apart from the training environments.

In the first part of Table 1, we list most of the previous works on the Room-to-Room dataset (Anderson et al., 2018b) and report the *success rate* under greedy decoding (i.e., without beam-search) on validation seen and validation unseen splits. The large absolute gaps (from $30.9\%$ to $9.7\%$) between the results of seen and unseen environments show that current neural agent models on R2R suffer from environment bias[2]. Besides Room-to-Room (R2R), we also analyze two newly-released indoor navigation datasets that were also collected from Matterport3D environments: Room-for-Room (R4R) (Jain et al., 2019) and Cooperative Vision-and-Dialog Navigation (CVDN) (Thomason et al., 2019b). As shown in the second and third parts of Table. 1, results drop significantly from seen to unseen environments (i.e., $26.9\%$ on R4R and $3.74$ on CVDN), indicating that agent models also suffer from the environment bias in these datasets. Lastly, we show the results (denoted as 'ours' in Table. 1) when the environment bias (reason analyzed in Sec. 5) is effectively eliminated by our learned semantic features (described in Sec. 6.3). As a result, the performance gaps are effectively decreased on all three datasets without changing the model and learning hyper-parameters, compared to our baselines (denoted as 'Our baseline') and previous works [3].

### 3.3 Environment Bias in Outdoor Navigation

Since the three indoor navigational datasets in previous sections are collected from the Matterport3D environments (Chang et al., 2017), in order to show that the environment bias is a general phe-

---

[2]Our work's aim is to both close the seen-unseen gap while also achieving competitive unseen results. Note that Anderson et al. (2019) also achieve 0% gap but at the trade-off of low unseen results. There is also another recent work by Ke et al. (2019) but they do not report val-seen results from non-beam-search methods.

[3]As for another major evaluation metric on the R4R dataset, Coverage weighted by Length Score (CLS), we also observe a similar phenomenon in performance gap; and our methods can also eliminate this gap from 19.2 to 1.5 and achieve competitive state-of-the-art unseen CLS results (34.7).

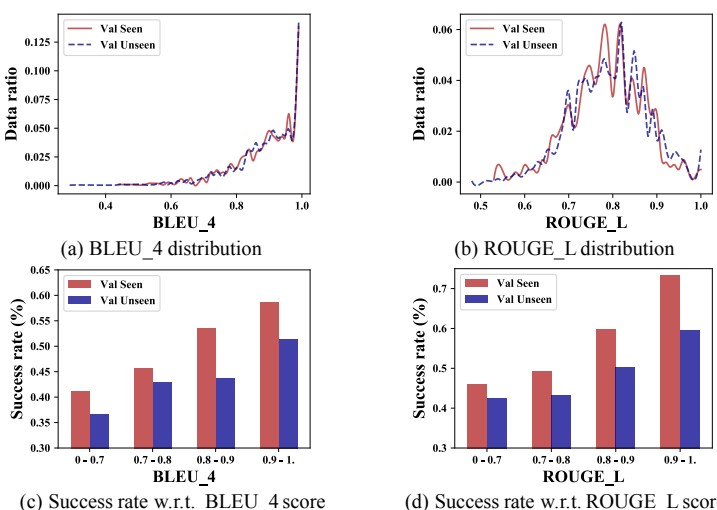

Figure 2: The language 'distance' distribution (defined by language scores) and its relationship to success rate.

nomenon also existing in other kinds of environments, we investigate the outdoor navigation task from Touchdown dataset (Chen et al., 2019c), whose environments are taken from New York City. In the original data splits of Touchdown, the environment is not specifically divided into seen and unseen and only involved one city. Thus the trained agent is only tested on the training environments (similar to validation seen split). To reveal the environment bias in Touchdown dataset, we split the city environment according to latitude and create two sub-environments: 'training' and 'unseen'. The data are then re-split into training, val-seen, and val-unseen, accordingly. We adapt our baseline R2R agent model with additional convolutional layers to fit this new task. As shown in the last part of Table. 1, when experimenting on the original data split, our baseline model achieves state-of-the-art results on the original 'dev' set and 'test' set, proving the validity of our model in this dataset. However, the results on our re-split data (denoted as 'Our baseline (seen/unseen split)') still show a big drop from the 'training' to the 'unseen' sub-environment (from $17.5\%$ to $5.3\%$), indicating that environment bias is a broad issue.

## 4    WHERE: THE EFFECT OF DIFFERENT TASK COMPONENTS

In Sec. 3, we showed that current neural agent models are biased towards the training environments on multiple vision-and-language navigation (VLN) datasets. In this section, our goal is to locate the component of VLN tasks which this environment bias is attributed to. As one of the early-released and well-explored datasets of VLN, Room-to-Room (R2R) dataset (Anderson et al., 2018b) is used as the diagnosing dataset in the experiments. We start by showing that two possible candidates, the natural language instructions and the underlying navigational graph, do not directly contribute to the environment bias. Then the effect of visual environments is analyzed in detail.

### 4.1    THE EFFECT OF NATURAL-LANGUAGE NAVIGATIONAL INSTRUCTIONS

A common hypothesis is that the navigational instructions for unseen environments (e.g., val unseen) are much different from the training environments (i.e., training and val seen) due to the different objects and layouts in new environments; and this lingual difference thus leads to the performance gap. In this section, we analyze the distributions of success rate with regard to the relationship between validation data's instructions and training instructions. In order to quantitatively evaluate this relationship, we define the 'distances' from a validating instruction to all training instructions as the phrase-matching metric. Suppose $x$ is a validating datum, $\mathbb{T}$ is the training set, and $\text{inst}(x)$ is the instruction of the datum $x$, we use ROUGE-L (Lin, 2004) and BLEU-4 (Papineni et al., 2002) to

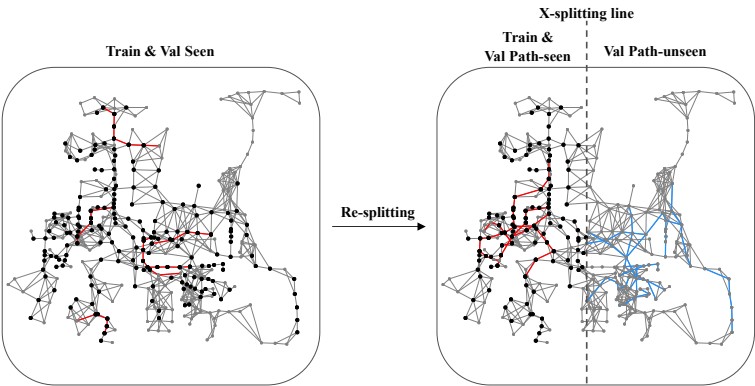

Figure 3: Graph split: left is original data and right is re-splitting data. Black vertices are viewpoints visited during training; red paths are val seen / val path-seen; blue paths are val path-unseen.

calculate this 'distance':

$$\text{dis}_{\text{ROUGE}}(x, \mathbb{T}) = \min_{t \in \mathbb{T}} \text{ROUGE-L}\left(\text{inst}(x), \text{inst}(t)\right) \tag{1}$$

$$\text{dis}_{\text{BLEU}}(x, \mathbb{T}) = \text{BLEU-4}\left(\text{inst}(x), \{\text{inst}(t)\}_{t \in \mathbb{T}}\right) \tag{2}$$

where we consider all the training instructions as references in calculating the BLEU-4 score.

We show the distributions of success rates and distances in Fig. 2. As opposed to the hypothesis, we do not observe a significant difference between the distributions of 'distances' (as shown in Fig. 2 (a, b)) on seen validation and unseen validation. For the success rate distributions (in Fig. 2(c,d)), the performance is better on instructions with smaller 'distances' (i.e., higher BLEU-4/ROUGE-L scores w.r.t. the training instructions) on both validation splits. However, comparing two splits, with the same 'distance' to training instructions, seen validation still significantly outperforms the unseen validation set on success rate, which implies the existence of other reasons rather than language attributed to this performance gap.

### 4.2 THE EFFECT OF UNDERLYING NAVIGATIONAL GRAPH

As shown in Fig. 3, an environment could be considered as its underlying navigational graph with visual information (as in Fig. 1). In order to test whether the agent model could overfit to these navigational graphs (and thus be biased towards training environments), we follow the experiments in Hu et al. (2019) to train the agent without visual information. Specifically, we mask out the ResNet features with zero vectors thus the agent could only make the decision based on the instructions and the navigational graph. With our baseline model, the success rate is $38.5\%$ on validation seen and $41.0\%$ on validation unseen in this setting, which is consistent with the finding in Hu et al. (2019). Besides showing the relatively good performance of unseen split without visual contents (similar to Thomason et al. (2019a) and Hu et al. (2019)), we also want to emphasize the low performance gap between seen and unseen environments ($2.5\%$ compared to the $> 10\%$ gap in usual). Hence, we claim that the underlying graph is not a dominant reason for the environment bias.

### 4.3 THE EFFECT OF VISUAL ENVIRONMENTS

To show how the visual environments affect the agent's performance, we analyze the results on unseen environments and in different spatial regions of the training environments. In order to give a detailed characterization of the effect of environments, we are going to reveal the spatial localities which are related to the agent's performance at three different levels:

- **Path-level Locality:** Agents are better at paths which intersect with the training paths.
- **Region-level Locality:** Agents are better in regions which are closer to the training data.
- **Environment-level Locality:** Agents perform better on training environments than on unseen environments.

Table 2: Results on our re-splitting data showing the path-level and environment-level localities.

| | Splitting Method | Train | Validation | | |
|---|---|---|---|---|---|
| | | | Path-seen | Path-unseen | Env-unseen |
| **Environments** | R2R | 61 | 56 | 0 | 11 |
| | X-split | 61 | 57 | 16 | 11 |
| | Z-split | 61 | 56 | 29 | 11 |
| **Number of Data** | R2R | 14,025 | 1,020 | 0 | 2,349 |
| | X-split | 11,631 | 1,230 | 1,098 | 2,349 |
| | Z-split | 10,894 | 867 | 2,324 | 2,349 |
| **Success Rate** | R2R | 88.3 | 56.1 | – | 47.5 |
| | X-split | 87.3 | 58.9 | 52.6 | 46.7 |
| | Z-split | 94.7 | 62.5 | 47.8 | 42.4 |

And the existence of these spatial locality inspires us to find the direct cause of the problem in Sec. 3.2. However, the original split of data is not fine-grained enough to separately reveal these spatial localities. To better illustrate this, we visualize the data from one environment of the Room-to-Room dataset in Fig. 3, where the vertices are viewpoints with visual information and edges are valid connections between viewpoints. The vertices highlighted with dark-black indicate the viewpoints which are used in training paths, and the red edges are the connections covered by original val-seen paths. As shown in Fig. 3, nearly all viewpoints in val-seen paths (vertices connected to red lines) are used as viewpoints in training data (vertices marked by dark-black). We thus cannot categorize the path-level and region-level localities. To bypass this, we propose a novel re-splitting method to create our diagnosis data splits.

**Structural Data Re-splitting** We employ two kinds of structural data splitting methods based on the horizontal or vertical coordinates, denoted as 'X-split' and 'Z-split', respectively. The 'Z-split' intuitively separates different floors in the houses and 'X-split' creates separate areas. When applying to the training environments in R2R dataset, we use one side of the splitting line (see the 'X-splitting line' Fig. 3) as the new training 'environment', and the other side as the path-unseen 'environment'. In addition to this split of environments, we also re-split the original training data and val-seen data while keeping the val-unseen data the same. The data paths across the splitting line are dropped. As shown in the right part of Fig. 3, we create three new data splits: training split, val-path-seen split, and val-path-unseen split. The edges covered by the new val-path-unseen split are highlighted in blue, while the color style of training split and val-path-seen split ('Black' for viewpoints in training and 'Red' for edges in val path-seen) are the same. Since the amount of original val-seen data are inadequate to fill two new validation sets (val path-seen and val path-unseen), we bring some (original) training data into our new validation splits. The overall statistics of original splits and our new splits are shown in Table 2.[4]

**Existence of Path-level and Environment-level Localities** For both splitting methods, we train our baseline model on the newly-split training set and evaluate on our three validation sets (denoted as 'X-split' or 'Z-split' rows in Table 2). The results of our baseline model on the original R2R (denoted as 'R2R' rows) splits are listed for comparison. As shown in Table. 2, the agent performs better on val path-seen than val path-unseen, which suggests that a **path-level locality** exists in current VLN agent models. Meanwhile, the results on val path-unseen are further higher than val env-unseen and it indicates the **environment-level locality** which is independent of the path-level locality.

**Existence of the Region-level Locality** To further demonstrate region-level locality, we study how the success rate changes in different regions of the environment with respect to their distances to the training data, which is similar to the analysis of language 'distance' in Sec. 4.1. We first calculate the point-by-point shortest paths using the Dijkstra's algorithm (Dijkstra, 1959), where the shortest distances between viewpoints $v$ and $v'$ are denoted as the graph distance $\mathrm{dis}_{\mathrm{GRAPH}}(v, v')$. Based on this graph distance, we define the viewpoint distance $\mathrm{dis}_{\mathrm{VIEWPOINT}}$ from a viewpoint $v$ to the training

---

[4]We only split the environments whose data contains substantial amount, thus make sure that the remaining training data is still adequate for training strong models.

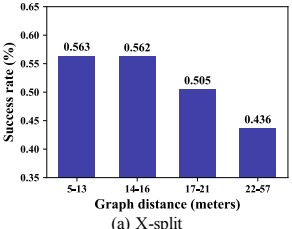 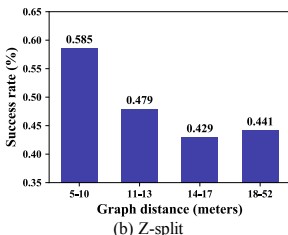

(a) X-split   (b) Z-split

Figure 4: The success rate declines as the path moves further from training regions.

data $\mathbb{T}$ as $v$'s minimal graph distance to a viewpoint $v'$ in training data. We then define the path distance $\text{dis}_{\text{PATH}}$ from a validating data $x$ to the whole training data $\mathbb{T}$ as the maximal viewpoint distance in the path of $x$:

$$\text{dis}_{\text{PATH}}(x, \mathbb{T}) = \max_{v \in \text{path}(x)} \text{dis}_{\text{VIEWPOINT}}(v, \mathbb{T}) \tag{3}$$

$$= \max_{v \in \text{path}(x)} \left\{ \min_{\substack{v' \in \text{path}(t) \\ \forall t \in \mathbb{T}}} \text{dis}_{\text{GRAPH}}(v, v') \right\} \tag{4}$$

We compute this path distance between paths in the env-seen validation set and training environments in our re-split data. As shown in Fig. 4, the success rate declines as the path moves further from the training environment on both re-splitting methods (i.e., 'X-split' and 'Z-split'). As conclusion, the closer the path to the training data, the higher the agent performance is, which suggests the existence of region-level locality.

## 5   WHY: WHAT INSIDE THE ENVIRONMENTS CONTRIBUTES TO THE BIAS?

In Sec. 4, we locate the cause of performance gap in visual environments by excluding other potential reasons and categorizing the spatial localities. However, there are still multiple possible aspects inside the environment which could lead to these spatial localities, e.g., the object layout convention and the room connections. The agent model could be biased towards the training environments by over-fitting or memorizing these environment-specific characteristics. In this section, we want to identify which aspect directly contributes to the bias and draw the following conclusion: the environment bias is attributed to low-level visual information carried by the ResNet features.

We first show an experiment that effectively decreases the gap between seen and unseen environments with minimal model modifications. We then clarify our conclusions based on the findings.

### 5.1   AN INVESTIGATION EXPERIMENT: IMAGENET LABELS AS VISUAL FEATURES

Suspecting that the over-fitting happens when the agent over-learns low-level features, we hope to find the replacement of ResNet 2048-features that contain minimal low-level information while preserving distinguishable visual contents. The most straightforward replacement is that instead of using mean-pooled features, we inject the frozen 1000-way classifying layer in ResNet pre-training, and use the probabilities of ImageNet labels as visual features. Shown as 'ImageNet' in Table. 3, the probability distribution almost closes the gap between seen and unseen. These results further constrain the reason of environment bias to the low-level ResNet features of image views. Combining with the findings of spatial localities, we suggest that environments (i.e., houses) and regions (i.e., rooms) usually have their own 'style'. Thus the same semantic label (captured by ImageNet-1000 features) has different visual appearances (captured by ResNet features) in different environments or regions. As a result, ImageNet-1000 features, in spite of being noisy, are not distracted by low-level visual appearance and could generalize to unseen environments, while ResNet features could not.

Although these ImageNet-1000 features decrease the performance gap, it has a disagreement with the VLN domain so that the validation unseen results of R4R and CVDN are slightly worse than baseline (and not much better for R2R). Hence it motivates us to find better semantic representations of environmental features that can both close the seen-unseen gap while also achieving state-of-the-art on unseen results (which we discuss next).

Table 3: Results showing that our semantic feature representations eliminate the performance gap in all three datasets.

| Task | Feature | | | Result | | |
| --- | --- | --- | --- | --- | --- | --- |
| | **Type** | **Name** | **Dim** | **Val Seen** | **Val Unseen** | **Abs Gap $\|\Delta\|$** |
| Room-to-Room | Baseline | ResNet NoDrop | $2,048$ | 54.5 | 38.2 | 16.3 |
| | Baseline | ResNet | $2,048$ | 56.1 | 47.5 | 8.6 |
| | Invesgation | ImageNet | $1,000$ | 47.1 | 48.2 | 1.1 |
| | Semantic | Detection | 152 | 55.9 | 50.0 | 5.9 |
| | Semantic | Ground Truth | 42 | 55.6 | 56.2 | 0.6 |
| | Semantic | Learned | 42 | 53.1 | 53.3 | 0.2 |
| R4R | Baseline | ResNet NoDrop | $2,048$ | 52.5 | 25.8 | 26.7 |
| | Baseline | ResNet | $2,048$ | 54.6 | 30.7 | 23.9 |
| | Investigation | ImageNet | $1,000$ | 28.7 | 28.9 | 0.2 |
| | Semantic | Detection | 152 | 48.8 | 32.0 | 16.8 |
| | Semantic | Ground Truth | 42 | 47.6 | 35.9 | 11.7 |
| | Semantic | Learned | 42 | 36.2 | 36.1 | 0.1 |
| CVDN | Baseline | ResNet NoDrop | $2,048$ | 5.88 | 2.14 | 3.74 |
| | Baseline | ResNet | $2,048$ | 5.97 | 2.23 | 3.74 |
| | Invesgation | ImageNet | $1,000$ | 3.22 | 2.08 | 1.14 |
| | Semantic | Detection | 152 | 3.34 | 2.08 | 1.26 |
| | Semantic | Ground Truth | 42 | 3.75 | 2.69 | 1.06 |
| | Semantic | Learned | 42 | 2.60 | 2.43 | 0.17 |

## 6 HOW: METHODOLOGY TO FIX THE ENVIRONMENT BIAS

In the previous section (Sec. 5), we found that the environment bias is related to the low-level visual features (i.e., 2048-dim ResNet features). Following the findings we observed in Sec. 5.1, we build our agent on the features which are more correlated to the VLN environmental semantics than the ImageNet label features in Sec. 5.1. We first demonstrate our baseline results on three VLN datasets and then explore the advanced semantic feature replacements. As shown in Table 3, these advanced semantic features could effectively reduce the performance gap between seen and unseen environments and improve the unseen results compared to our strong baselines. The effectiveness of these semantic features supports our explanation of the environment bias in Sec. 5 and also suggests that future work in VLN tasks should think about such generalization issues.

### 6.1 BASELINE

In our baseline model, following the previous works we use the standard ResNet features as the representation of environments (Anderson et al., 2018b; Jain et al., 2019; Thomason et al., 2019b). These features come from the mean-pooled layer after the final convolutional layer of ResNet-152 (He et al., 2016) pre-trained on ImageNet (Russakovsky et al., 2015). As shown in 'Baseline'[5] rows of Table. 3, val-seen results are significantly higher than val-unseen results in all three datasets. Note that our baseline method takes the 'feature dropout' technique demonstrated in Tan et al. (2019) (without back translation): the ResNet features are randomly masked by zero before used as inputs of the agent. Without this 'feature dropout' (denoted as 'ResNet NoDrop' in Table. 3), the gaps will increase in R2R and R4R, which suggests that this 'feature dropout' technique also helps to eliminate the low-level visual information over-fitting as we discussed in Sec. 5. However, the performance gap is still large, which leads us to the following discussions of semantic features.

### 6.2 DETECTED OBJECTS AREAS

During navigation, the objects in the environments are crucial since their matchings with the instruction often indicate the locations that can guide the agent, thus object detection results of the environments can provide relevant semantic information. In our work, we utilize the detection information generated by Faster R-CNN (Ren et al., 2015) to create the feature representations. Comparing to

---

[5]We use our baseline agent model (in Sec. 3.1) for R2R and R4R. For CVDN, we take the official baseline code in https://github.com/mmurray/cvdn.

ImageNet-1000 features (Sec. 5.1), these detection features include more environmental information since the viewing images in VLN usually contain multiple objects. Instead of directly using classification probabilities of the labels from ResNet and different from the approach in Hu et al. (2019) who utilized the embeddings of detected labels, we design our detection features $f_{\text{DETECT}}$ of each image view as the sum of the areas of detected objects weighted by detection confidence:

$$f_{\text{DETECT}}=[a_{c_1}, a_{c_2}, \ldots, a_{c_n}]; \qquad a_{c_i}=\sum_{\text{obj is } c_i} \text{Area(obj)} \cdot \text{Conf(obj)} \qquad (5)$$

where the $c_i$ and $a_{c_i}$ are the label and feature of each detected object, $\text{Area}(*)$ and $\text{Conf}(*)$ are the area and confidence of each object. For implementation details, we use the Faster R-CNN (Ren et al., 2015) trained on Visual Genome (Krishna et al., 2017) provided in Bottom-Up Attention (Anderson et al., 2018a). To eliminate the labels irrelevant to VLN task, we calculate the total areas of each detection object among all environments and pick the labels that take up a relatively large proportion of the environments, creating features of dimension 152.[6] Denoted as 'Detection' in Table 3, the performance gap is diminished with these detection features compared to baselines in all three datasets, indicating that changing the features to a higher semantic level has a positive effect on alleviating the environment bias. Meanwhile, the improvement of unseen validation results on R2R an R4R datasets suggests the better efficiency in the VLN task than the ImageNet labels.

### 6.3 SEMANTIC SEGMENTATION

Although the detection features can provide adequate semantic information for the agent to achieve comparable results as the baseline model, they do not fully utilize the visual information where the content left over from detection may contain useful knowledge for navigation. A better semantic representation is the semantic segmentation, which segments each view image on the pixel level and gives the label to each segment region, allowing us to utilize the semantics from the entire environment. Matterport3D (Chang et al., 2017) dataset provides the labeled semantic segmentation information of every scene and we take the rendered images from Tan et al. (2019)[7]. A comparison example of RGB images and semantic views is available in the Appendix. Since the semantic segmentation images are fine-grained and blurry in boundaries, we follow the design of detection features, using the areas of semantic classes in each image view as the semantic features (confidence is excluded since semantic segmentation does not provide this value). The areas are normalized to $[0, 1]$ by dividing the area of the whole image region. We first assume that the semantic information is provided as additional environmental information and the results of the model using the ground truth semantic areas are shown in the 'ground truth' rows in Table. 3. We next study the situation where the semantic information is not available in testing environments thus the information needs to be learned from training environments. Thus we train a separate multi-layer perceptron to predict the areas of these semantic classes (details in Appendix), and the results of the model with these predicted semantics as features are shown in 'learned'. As shown in Table. 3, both 'ground truth' and 'learned' semantic representations bring the performance of seen and unseen closer comparing to the baseline model, and the smallest performance gaps come from learned semantic segmentation features in all three datasets. The highest validation unseen success rates among all the proposed feature representations are also produced by semantic segmentation features, 'learned' semantic for R4R and 'ground truth' semantic for R2R and CVDN. Overall, among all the semantic representations we have explored, the semantic segmentation features are most effective in eliminating the environment bias.

## 7 CONCLUSION

In this paper, we focus on studying the performance gap between seen and unseen environments widely observed in vision-and-language navigation (VLN) tasks, trying to find where and why this environment bias exists and provide possible initial solutions. By designing the diagnosis experiments of environment re-splitting and feature replacement, we locate the environment bias to be in the low-level visual appearance; and we discuss semantic features that decrease the performance gap in three VLN datasets and achieve state-of-the-art results.

---

[6]Note that this detection feature dimension 152 coincidentally is the same as the number of layers in ResNet, but there is no correlation.

[7]The rendered semantic views are downloaded from https://github.com/airsplay/R2R-EnvDrop.

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

# A APPENDIX

RGB Images 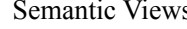 Semantic Views

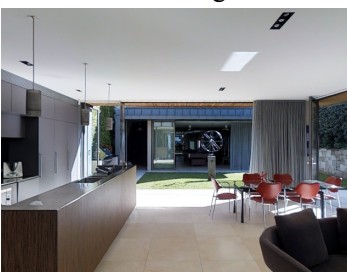 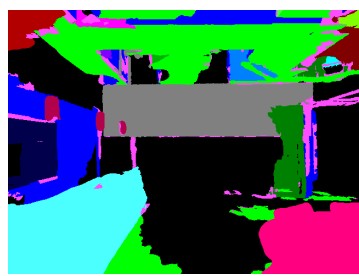

Figure 5: Comparisons between RGB images and their semantic views.

## A.1 EXAMPLES OF RGB IMAGES AND SEMANTIC VIEWS

In Fig. 5, we show a rendered semantic view from Tan et al. (2019) and its original RGB image. Different colors indicate different semantic segmentation areas and 40 semantic labels are considered in the Matterport3D dataset Chang et al. (2017).

## A.2 DETAILS OF 'LEARNED' SEMANTIC TRAINING

We use a multi-layer perceptron over the ResNet features to generate the 'learned' semantic features. The multi-layer perceptron includes three fully-connected layers with $\mathrm{ReLU}$ activation on the outputs of the first two layers. The input is the 2048-dim ResNet feature $f$ of each image view. The hidden sizes of the first two layers are 512 and 128. The final layer will output the 42-dim semantic feature $y$ that represents the areas of each semantic class. After the linear layers, we use the sigmoid function $\sigma$ to convert the output to the ratio of areas.

$$x_1 = \mathrm{ReLU}(A_1 f + b_1) \tag{6}$$
$$x_2 = \mathrm{ReLU}(A_2 x_1 + b_2) \tag{7}$$
$$y = \sigma(A_3 x_2 + b_3) \tag{8}$$

The model is trained with ground truth semantic areas $y_{\text{AREA}}$ (normalized to $[0, 1]$) and only the views in training environments are used in training. We minimize the binary cross-entropy loss between the ground truth areas $\{y_i^*\}$ and the predicted areas $\{y_i\}$, where $i$ indicate the $i$-th semantic class.

$$\mathcal{L} = -\sum_i \left( y_i^* \log y_i + (1 - y_i^*) \log (1 - y_i) \right) \tag{9}$$

Dropout layers with a probability of $0.5$ are added between fully-connected layers while training. The sigmoid function $\sigma$ and the cross-entropy loss are combined to improve numerical stability.

After the model is fitted, we freeze the weight and use it to predict the semantic features of all seen and unseen environments (i.e., environments for training, val-seen, and val-unseen data). The predicted features are then used as the input of our neural agent model for different datasets (i.e., R2R, R4R, and CVDN), and the neural agent models are the same except we change the input dimension from $2048$ (the dimension of ResNet features) to $42$ (the number of semantic classes).

