# OpenReview forum: "Diagnosing the Environment Bias in Vision-and-Language Navigation"
_ICLR.cc/2020/Conference — Reject_

### Official Review · AnonReviewer1 · 2019-10-22
**Official Blind Review #1**

**Rating:** 6

**Review:**

This paper aims to identify the primary source of transfer error in vision&language navigation tasks in unseen environments. The authors tease apart the contributions of the out-of-distribution severity of language instructions, navigation graph (environmental structure), and visual features, and conclude that visual differences are the primary form in which unseen environments are out of distribution. They show that using ImageNet class scores as visual features results in significantly less transfer gap than using low-level visual features themselves. Experiments then show that semantic-level features dramatically reduce the transfer gap, although at a cost of absolute performance.

I recommend this paper for acceptance; my decision is based on the thorough analysis of the ultimate cause of a recurring problem in this field.

These results, if shown to hold across a significant number of datasets and tasks, would significantly change the focus of research in this field toward a focus on robust high-level visual representations (as opposed to e.g. better spatial awareness or better language understanding). This work represents an important step in this direction.

The description of the 'learned' features in 6.3 could use more elaboration. Since it is the best performing approach by a large margin (as measured by transfer gap), it should probably get more than one sentence. In particular, what do the authors mean by "train a separate multi-layer perceptron to predict the areas of these semantic labels"? Does that mean the predicted pixel-level semantic segmentation map is used as input to the navigating agent? Or is it an auxiliary task for representation learning? etc. This should be clarified.

I anticipate this paper to significantly influence future work in this area.

--------

After discussing with the reviewers about the methodological issue of the validation set, I have lowered my score to a weak accept, but I think this paper should still be published.

**Experience Assessment:**

I have published one or two papers in this area.

**Review Assessment: Checking Correctness Of Derivations And Theory:**

N/A

**Review Assessment: Checking Correctness Of Experiments:**

I assessed the sensibility of the experiments.

**Review Assessment: Thoroughness In Paper Reading:**

I read the paper at least twice and used my best judgement in assessing the paper.

---

> ### Author Response · Authors · 2019-11-14
> **Response to Blind Review #1:**
>
> We thank the reviewer for appreciating our detailed analysis and contributions to the community. To make sure the reproducibility of this paper and lead to future research in VLN, we will publicly release the features and our code when the anonymous period ends. Furthermore, we will also provide a unified script that could convert most existing Github projects with Matterport3D environments to use the semantic features. Some example projects are here (sorted by time):
> https://github.com/peteanderson80/Matterport3DSimulator
> https://github.com/ronghanghu/speaker_follower
> https://github.com/chihyaoma/selfmonitoring-agent
> https://github.com/chihyaoma/regretful-agent
> https://github.com/Kelym/FAST
> https://github.com/airsplay/R2R-EnvDrop
> https://github.com/mmurray/cvdn
> After running the script, all the code above will run with its own model but with our provided features.
>
> - Details of learned features:
> Thanks for the suggestions. We have clarified the ‘learned semantic features’ in the Appendix of the revised pdf version. The multi-layer perceptron is a separate module from the neural agent model, which has three FC layers with ReLU activation, projecting 2048-ResNet features to 42-dimension semantic features. We trained it to directly predict the area of each semantic class (as the “ground truth” semantic features) instead of building the semantic segmentation first. Then the model is frozen and used to generate the semantic features of seen and unseen environments, which will be the input of the navigational agent in replace of ResNet features. We do not consider it as an auxiliary task but it would be a good future direction to work on.

---

### Official Review · AnonReviewer3 · 2019-10-22
**Official Blind Review #3**

**Rating:** 3

**Review:**

This paper has two main contributions. First, the authors perform an extensive study to understand the source of what they refer to as 'environment bias', which manifests itself as a gap in performance between environments used for training and unseen environments used for validation. The authors conclude that of the three sources of information provided to the agent (the natural language instruction, the graph structure of the environment, and the RGB image), the RGB image is the primary source of the overfitting. The second contribution is to use semantic information, compact statistics derived from (1) detected objects and (2) semantic segmentation, to replace the RGB image and provide input to the system in a way that maintains state-of-the-art performance but shrinks the performance gap between the seen and unseen data.

This paper has some pretty exhaustive treatment diagnosing the source of the agent's 'environment bias' (which, as I discuss below, I believe is more accurately referred to as 'overfitting') in Sec. 4. To me, this is this highlight of the paper, and some interesting work; the investigation of the behavior of the system is interesting and informative. It provides a framework for thinking about how to diagnose this behavior and identify its source. The authors use this rather extensive study to motivate the need for new features (semantic features) to replace the RGB image that their investigation finds is where much of this 'environment bias' is located. Unfortunately, it is here that the paper falls flat. The authors proposal methods perform nominally better on the tasks being investigated, but much of the latter portion of the paper continues to focus on the 'improvement' in the metric they use to diagnose the 'bias'. As I mention below, the metric for success on these tasks is performance on the unseen data, and, though an improvement on their 'bias' metric is good anecdotal evidence their proposed methods are doing what they think, the improvements in this metric are largely due to a nontrivial decrease in performance on the training data. Ultimately, this is not a compelling reason to prefer their method. I go into more details below about where I think some of the other portions of the paper could be improved and include suggestions for improvement.

High-level comments:
- I am uncertain that 'bias' is the right word to describe the effect under study. In my experience, environment bias (or, more generally, dataset bias) usually implies that the training and test sets (or some subset of the data) are distinct in some way, that they are drawn from different distributions. The learning system cannot identify these differences without access to the test set, resulting in poor performance on the 'unseen' data. In the scenario presented here, the environments are selected to be in the train/test/validation sets at random. As such, the behavior described here is probably more appropriately described as 'overfitting'. The shift in terminology is not an insignificant change, because using 'bias' to describe the problem incorrectly suggests that the data collection procedure is to blame, rather than a lack of data or an overparamatrized learning strategy; I imagine that more data in the training set (if it existed) could help to reduce the gap in performance the paper is concerned with. That being said, I imagine some language changes could be done to remedy this.
- Perhaps the biggest problem with the paper as written is that I am not convinced that the 'performance gap' between the seen and unseen data is a metric I should want to optimize. This metric is instructive for diagnosing which component of the model the overfitting is coming from, and Sec. 4 (devoted to a study of this effect) is an interesting study as a result. However, beyond this investigation, reducing the gap between these two is not a compelling objective; ultimately, it is the raw performance on the unseen data that matters most. The paper is written in a way that very heavily emphasizes the 'performance gap' metric, which gets in the way of its otherwise interesting discussion diagnosing the source of overfitting and some 'strong' results on the tasks of interest. The criteria should be used to motivate newer approaches, rather than the metric we should value for its adoption. This narrative challenge is the most important reason I cannot recommend this paper in its current state.
- Using semantic segmentation, rather than the RBG image, as input seems like a good idea, and the authors do a good job of motivating the use of semantics (which should show better generalization performance) than a raw image. However, the implementation in Sec. 6.3 raises a few questions. First (and perhaps least important) is that 6.3 is missing some implementation details. In this section, the authors mention that 'a multilayer perceptron is used' but do not provide any training or structure details; these details should be included in an appendix. More important is the rather significant decrease in performance on the seen data (11% absolute) when switching to the learned method. Though the performance on the unseen data does not change much, it raises some concerns about the generalizability of the learning approach they have used: in an ideal world with infinite training data, the network would perfectly accurately reproduce the ground truth results, and there should be no difference between the two. Consequently, the authors should comment on the discrepancy between the two and the limits of the learned approach, which I worry may limit its efficacy if more training data were added.

Smaller comments:
- I do not fully understand why the 'Touchdown' environment was included in Table 1, since the learned-semantic agent proposed in the paper was not evaluated. The remainder of the experiments are sufficient to convince the reader that this gap exists, and I would recommend either evaluating against the proposed technique or removing this task from the paper.
- Figure captions should be more 'self-contained'. Right now, they describe only what is shown in the figure. They should also describe what I, as a reader, should take away or learn from the figure. This is not always necessary, but in my experience improves readability, so that the reader does not need to return to the body of the text to understand.
- The use of a multilayer perceptron for the Semantic Segmentation learned features, trained from scratch, stands out as a strange choice, when there are many open source implementations for semantic segmentation exist and could be fine-tuned for this task; a complete investigation (which may be out of scope for the rebuttal period) may require evaluating performance of one of these systems.

**Experience Assessment:**

I have read many papers in this area.

**Review Assessment: Checking Correctness Of Derivations And Theory:**

I assessed the sensibility of the derivations and theory.

**Review Assessment: Checking Correctness Of Experiments:**

I assessed the sensibility of the experiments.

**Review Assessment: Thoroughness In Paper Reading:**

I read the paper thoroughly.

---

> ### Author Response · Authors · 2019-11-14
> **Response to Blind Review #3 (part 1 of 2):**
>
> We thank the reviewer for appreciating our informative investigation of the system behavior and comprehensive diagnosis experiments.
>
> - Performance Gap
> Thanks for your thoughtful questions regarding the ‘performance gap’. We would like to answer these broad questions point-by-point.
>
> 1. Val unseen is the major evaluation set.
> As we clarified in the abstract and Sec. 1, performance on unseen environment has mainly been evaluated in instruction-guided vision-and-language navigation because these step-by-step instructions are too detailed to navigate seen environments. For example, suppose that I am in my home (which is an example of seen environment that I am already familiar with) and someone is asking for my help in the kitchen. They might say “Please come to the kitchen” instead of “Please go outside the bedroom, turn left and face towards the table. Go across the table and enter the door of the kitchen to your right”.  Overall, we would like to control an embodied agent with short, informative instructions in seen environments, and thus the instruction-guided VLN task has limited applications, so most of the existing VLN tasks are only compared on val unseen.
>
> 2. ‘Performance gap’ measures the generalizability to unseen environments.
> Since we mainly care about the agents’ performance in unseen environments, metrics regarding val seen are all considered as diagnosis metrics. The val-seen success rate resembles the training accuracy in other tasks (e.g., image classification). While the training accuracy shows some characteristics of the learning method, it is uncorrelated to model's actual performance: 100% training accuracy and 0% validation accuracy is still considered as a bad result. Thus, the difference between the training accuracy and validation accuracy is more informative, which measures the ‘generalizability’ of the model. Similarly, the ‘performance gap’ in VLN tasks between two validation sets measures the ‘generalizability’ from training environments to unseen testing environment.
> However, different from the ‘overfitting’ in deep learning which seems to be a universal ‘benign’ (Zhang et.al., ICLR 2017) issue, the environment ‘bias’ seems not to be the same case: we show that it could be effectively reduced by semantic features while improving the val-unseen results (shown in Table 3) without changes in model architecture or training procedure. This is one of the main reasons that we take the word ‘bias’ instead of ‘overfitting’ in our paper.
>
> [Zhang et.al., ICLR 2017]  Zhang, C., Bengio, S., Hardt, M., Recht, B., & Vinyals, O. Understanding deep learning requires rethinking generalization. ICLR 2017.
>
> 3.  Diagnosis experiments are designed to locate the reason for this performance gap.
> Since the meaning of the performance gap arises naturally, the metric is not specifically designed for our diagnosis experiments. We actually conduct experiments to recover the reasons behind the problem of the large performance gap. Therefore, we believe that this is different from the arguments: ‘but much of the latter portion of the paper continues to focus on the 'improvement' in the metric they use to diagnose the 'bias’’ and ‘This metric is instructive for diagnosing which component of the model the overfitting is coming from’.
>
> 4. Our methods still optimize the val-unseen results while taking the generalization into consideration.
> We agree that ‘the raw performance on val unseen data matters the most’, and this is what we pursued in the paper. As shown in Table 3, a consistent increase on val unseen could be observed. It means that semantic features could improve test results (i.e., val unseen results) while improving the neural model’s generalization. The paper also clarified that our model ‘achieves strong results on testing unseen environments’ in the Abstract, Sec. 1 Introduction, Sec. 6 Methodology, and Sec. 7 Conclusion. Since the main purpose of this paper is to show the reasons and potential solutions to the performance gap, we thus also emphasize the effectiveness of our method in improving the generalizability.
>
> Meanwhile, most regularization methods on preventing overfitting (e.g., dropout and weight regularization) would increase the testing results by hurting the training accuracy/loss. The same thing happens here where the val-unseen success rate increases and val-seen success rate decreases.
>
> 5. The criteria should be used to motivate newer approaches.
> We believe that these experiments and results regarding the performance gap will lead to new methods in VLN tasks. As mentioned by Reviewer #1, they ‘would significantly change the focus in this field toward a focus on robust high-level visual representations (as opposed to e.g. better spatial awareness or better language understanding)’. Our semantic-feature approaches in Sec. 6 are initial attempts in this direction which improve the val-unseen results following the findings in the paper.

---

> > ### Author Response · Authors · 2019-11-14
> > **Response to Blind Review #3 (part 2 of 2):**
> >
> > - About the word 'bias':
> > Thanks for the suggestions. As you mentioned, “environment bias usually implies that the training and test sets (or some subset of the data) are distinct in some way, that they are drawn from different distributions”; and in fact this is the case for Room2Room dataset, where the val unseen is created from completely different/unused house environments that have not been included in training/val-seen at all (see Sec. 4.4 and Figure 9 in Anderson et al. 2018b). Each individual environment is more like a small ‘domain’ with its own distribution to generate the views, the object layouts, and connectivity graphs (please see Fig. 9 in Anderson et al. 2018b). Also, note that we are not blaming the data collection at all. In fact, this 'bias' is naturally (deliberately) conveyed by the way that the validation sets are created in the R2R/VLN papers (with one set using training environments and the other using unique unseen environments), and created as an important factor to test the models’ generalization ability in follow-up works. Even with more training data (distinct from the unseen environments), these two kinds of environments will still be distinctive to each other, thus the seen vs. unseen bias will still exist (e.g., see Sec. 7.2 and Fig. 5 of Tan et al., 2019).
> >
> > Besides these above reasons, we use the word ‘bias’ instead of ‘overfitting’ to avoid misleading the reader:
> > 1. As we mentioned in the above response “Performance-Gap 2”, ‘overfitting’ is mostly an unavoidable phenomenon in training deep neural networks but it is not the same case to the environment ‘bias’.
> > 2. ‘Overfitting’ is between the training data and the validation data while environment ‘bias’ is between two validation sets: val seen and val unseen. The overfitting also exists in our neural agent models where it could achieve 80~90% on training data (not val seen).
> >
> >
> > - Details for learned semantic features:
> > Thanks for the suggestions. We have added implementation details in the Appendix of the updated version. The multi-layer perceptron is three FC layers with ReLU activation, projecting 2048-ResNet features to 42-dimension semantic features.
> >
> > The drop of val-seen results is a combination of two reasons. Firstly, we tuned the hyperparameter of the multi-layer perceptron to prevent overfitting the ground truth semantics features in training environments, thus the learned semantic features on the training environment (i.e., the val-seen data) are different from ground truth. Secondly, we use predicted features as the input for both training environments and unseen testing environments, to keep the feature distributions in both environments consistent. Overall, the val-seen results with the learned semantic features are decreased compared to ground truth semantic features. However, this feature-learning setup gives the highest val-unseen results in our experiments compared to advanced models which could overfit the training environments.
> >
> > - Showing performance gap on Touchdown:
> > As we mentioned in Sec. 3.3, all the other three datasets (i.e., R2R, R4R, and CVDN) are collected from the same in-door environments, Matterport3D (https://github.com/niessner/Matterport). Thus diagnosis experiments on these three datasets possibly reveal a characteristic of the specific environments (Matterport3D) instead of showing a characteristic of the general VLN task. To be best served as a comprehensive analysis of VLN tasks, we thus show the performance gap on one more navigation dataset Touchdown, which is an outdoor navigation task collected from Google Maps and provides very different environments from Matterport3D. Hence, by showing that current neural agent models on Touchdown dataset are still biased towards their training environments (regions in the city), it makes it more convincing that the performance gap is a universal phenomenon in VLN tasks.
> >
> > Thanks for the suggestion to apply semantic features to Touchdown. However, the raw RGB images (as mentioned in the footnote of Sec. 1 and please see https://github.com/lil-lab/touchdown for details) have not been released yet.
> >
> > - Figure captions:
> > Thanks for the advice about the figure caption. We have resolved this in the updated version.
> >
> > - Semantic segmentation training:
> > Thanks for the advice of semantic segmentation training. Our ground-truth semantic features are designed to be the areas of each semantic class (see Appendix for details). The purpose of our paper is to demonstrate the possibility of utilizing semantic features, we thus use simple MLP to regress these features instead of training a segmenter and then calculating the areas. Moreover, to fully take advantage of the semantic segmentation, the neural agent model needs modifications to adapt this new input and the noisy semantic segmentation (as shown in Fig. 5) need to be incorporated. That is beyond the scope of this analysis paper. We will explore these methods in future work with advanced methods.

---

> > > ### Comment · AnonReviewer3 · 2019-11-15
> > > **Clarification question about hyperparameter tuning**
> > >
> > > Thank you for taking the time to prepare such thorough responses to my comments. I am still considering some of the earlier comments, though I have a clarification question about your comment: "The drop of val-seen results is a combination..." Could you please clarify (in the comments) how the hyper-parameter tuning was performed? My assumption was that you trained on the training data, and tuned the parameters to maximize performance on Val Seen (and not Val Unseen); is this correct?

---

> > > > ### Author Response · Authors · 2019-11-15
> > > > **Hyper-parameters of the MLP are tuned based on val unseen**
> > > >
> > > > Thanks! We tune the hyper-parameters of the MLP (which learns the semantic features) w.r.t the loss on val-unseen environments for two reasons:
> > > > 1. The MLP generates learned semantic features as the input of the neural agent model. The agent models are tuned on the val-unseen environments to help build generalizable agents, we thus also tuned the MLP for this purpose since it could be considered as a part of the neural agent model.
> > > > 2. Val-seen and training data of the MLP (semantic features) are from the same set of training environments.
> > > >
> > > > By the way, tuning an additional module (such as an MLP in our paper) based on val unseen is also adopted by multiple previous works, e.g., [Fried et al., NeurIPS 2018] and [Tan et al., NAACL 2019] tune the performance of an additional speaker module on the val-unseen set. This strategy is considered as fairly comparable with other methods

---

> > > > > ### Comment · AnonReviewer3 · 2019-11-15
> > > > > **Further clarification about training and hyperparameter tuning**
> > > > >
> > > > > Thanks for your quick reply! Both of the papers you refer to in the previous comment, [Fried et al., NeurIPS 2018] and [Tan et al., NAACL 2019], include performance on an independent Test set (also from unseen environments). This is so that it is clear that the performance of the network is not being fine-tuned (via the hyperparameters) to the validation datasets, which are being used during construction of the network. Do you have results on such a Test set available? If not, this is a potential methodological concern.

---

> > > > > > ### Author Response · Authors · 2019-11-15
> > > > > > **About test-unseen results**
> > > > > >
> > > > > > Thanks for your quick reply as well! The test-unseen split (named as 'test' in some materials) is hidden and the test-unseen environments are different from the val-unseen environments. To avoid ‘fine-tuning’ on the hidden test set, the testing server only allows very submission. Thus previous analysis papers [Thomason et al., NAACL 2019a] (Table 1&2 in https://arxiv.org/abs/1811.00613) and [Hu et al., ACL 2019] (Table 1, 2, &3 in https://arxiv.org/pdf/1906.00347.pdf) only show val-seen and val-unseen results without testing the model performance on test-unseen data because analysis papers usually have multiple results. We do not test and show the test-unseen results in our paper for the same reason.
> > > > > >
> > > > > > In order to address the ‘potential methodological concern’ as you pointed out, since the best way to verify the validity of our ‘learned semantic features’ is comparing its results with the ‘baseline ResNet’ results on test unseen, we submit the predictions of our neural agent model with ResNet features (‘Baseline-ResNet’) and ‘learned semantic features’ (‘Semantic-Learned’) to the test server. The success rates are 46.0% and 51.2%. The 5.2% improvement on this ‘true-unseen’ test set shows that our learned semantic features significantly outperforms the original ResNet features.

---

### Official Review · AnonReviewer2 · 2019-10-24
**Official Blind Review #2**

**Rating:** 6

**Review:**

Summary: This paper provides a thorough analysis of why vision-language navigation (VLN) models fail when transferred to unseen environments. The authors enumerate potential sources of the failure--namely, the language, the semantic map, and the visual features--and show that the visual features are most clearly to blame for the failures. Specifically, they show that by removing the low-level visual features (e.g. the fc17 or similar) and replacing with various higher-level representations (e.g. the softmax layer of the pretrained CNN, or the output of a semantic segmentation system) dramatically improves generalization without a meaningful drop in absolute performance.

Evaluation: The paper is easy to follow and interesting. Some results presented have been show previously (e.g. that removing visual features doesn't drastically hurt performance of VLN models) but overall, the paper presents the results in a clear and thorough manner that will be beneficial to the community. A few small questions/comments below.

* I am confused by how you compute BLEU in Section 4.1. You say you compute corpus BLEU but Eq. 2 suggests you compute the BLEU for a single instruction against a set of training instructions. I think corpus BLEU is usually corpus vs. corpus (e.g. all generated sentences vs. all reference sentences) not one generated sentence against all reference sentences. Is this right? It also seems odd that your BLEU scores are distributed the way they are (Fig. 2). Can you explain why you did this the way you did?
* nit: Sec. 5 heading. Your grammar is backwards. The question you are trying to express is "bias is attributed to what" not "what is attributed to bias". So heading should be "to what inside the environments is bias attributed" (which is admittedly a clunky title)
* another nit: "suggest a surprising conclusion: the environment bias is attributed to low-level visual information carried by the ResNet features." --> idk that this is that surprising, it was kind of natural given the result that removing visual features entirely doesn't hurt performance and helps generalization. So maybe rephrase this sentence.



**Experience Assessment:**

I have read many papers in this area.

**Review Assessment: Checking Correctness Of Derivations And Theory:**

I assessed the sensibility of the derivations and theory.

**Review Assessment: Checking Correctness Of Experiments:**

I assessed the sensibility of the experiments.

**Review Assessment: Thoroughness In Paper Reading:**

I read the paper at least twice and used my best judgement in assessing the paper.

---

> ### Author Response · Authors · 2019-11-14
> **Response to Blind Review #2:**
>
> We thank the reviewer for appreciating our thorough analysis and contribution to the community.
>
> - BLEU score:
> Thanks for pointing it out. We use the term ‘corpus-level BLEU score’ to indicate that the references come from the whole training corpus instead of instructions in related environments. Therefore, it is indeed equivalent to the ‘sentence-level BLEU score’. We are sorry for the misleading and have modified it to “BLEU-4 score” in the updated pdf.
>
> - Nits:
> Thanks for the writing suggestions. We have changed the heading and rephrased the sentence in our updated version.
>
> - Removing visual features:
> Our work focuses on giving a comprehensive study of the factors that cause the environment bias. We thus wrote the paper in a way so as to not to take credits for the experiments in the papers “Are you looking” (Hu et.al, ACL 2019) and “Shifting the Baseline” (Thomason et.al, NAACL 2019a), who show that removing visual features does not drastically hurt model’s unseen performance. We instead demonstrate results from a different perspective of generalization: the seen-unseen performance gap is significantly dropped, which supports our hypothesis to eliminate the navigational graph as the dominant reason.

---

### Author Response · Authors · 2020-01-23
**Final comments and clarification**

We are the authors of the submission “Diagnosing the Environment Bias in Vision-and-Language Navigation”. First, we would like to thank PC/AC’s effort for managing the reviewing process of our paper and all the reviewers’ efforts and suggestions. However, just to clarify, we address below the concerns/misunderstanding raised by a reviewer regarding tuning the models on the “unseen-validation” data and not including “test” results in the paper:

(1) First, to address the concern about the performance on the test set, note that due to the #submissions limit of the testing server (https://evalai.cloudcv.org/web/challenges/challenge-page/97/overview), which limits every paper to test the model only a few times, hence all existing analysis papers in this area ([Thomason et al., NAACL 2019a] and [Hu et al., ACL 2019]) did not report the test results since analysis papers usually need multiple approaches to be tested and compared. For the same reason, we didn’t report test numbers in our original paper (and instead we use the val-unseen set, which could be viewed as a public test set that allows checking the generalization to new unseen environments that are separate from the training and val-seen environments). And note that all our learned and non-learned semantic-feature methods are fairly tuned on the same val-unseen.

Having said that, during the rebuttal period, we already showed the result of the selected methods (i.e., learned-semantic-feature method) on the test set (which in turn again has completely different environments from val-unseen) in the reply to AnonReviewer3 (https://openreview.net/forum?id=S1eYKlrYvr&noteId=BylcucJ2sS) and the generalization improvement on test is still substantial, showing the effectiveness of the initial proposed solution.

(2) Second, tuning the models on the validation-unseen set is a fair and comparable approach in the tasks we are studying, which is adopted by the majority of the previous works. We have the confirmation from the author of the original Room-to-Room dataset in May 2018 to “choose the best model using the val-unseen split”.

(3) Third, to further prove the validity of our ‘learned-semantic’ method, we re-train our MLP (which learns the semantic features) and tune it on val-seen (as opposed to the original version of tuning on val-unseen). Specifically, we randomly split the training environments and use 10 scenes as the new validation set and the rest as the new training set. The results with the re-trained features are consistent with our original conclusion (shown as the table below), that the ‘learned semantic’ features decrease the performance gaps while providing good val-unseen results. It also indicates that the tuning method used in our paper is not the major factor that contributes to our model’s performance.

Dataset  method                           Val-seen  Val-unseen  Gap
----------------------------------------------------------------------------------
R2R       Our baseline                     56.1        47.5              8.6
              Old learned-semantic     53.1        53.3              0.2
              New learned-semantic    52.6        53.3              0.7
----------------------------------------------------------------------------------
R4R       Our baseline                   54.6        30.7              23.9
              Old learned-semantic     36.2        36.1              0.1
              New learned-semantic    38.0        34.3              3.7
---------------------------------------------------------------------------------
CVDN    (**updated with panoramic-view)
              Our baseline                   6.60        3.05              3.55
              Old learned-semantic     5.74        4.31              1.43
              New learned-semantic    5.82        4.42              1.41

(4) Lastly, our paper is primarily focusing on the diagnosis of the ‘performance gap’ and the generalization issue in the vision-and-language navigation area and not trying to beat state-of-the-art or play the leaderboard game. In Sec. 3 of our paper, we discussed our observations of ‘performance gap’ in VLN area, and in Sec. 4 and 5, we designed elaborated diagnosis experiments to analyze where this ‘bias’ is located and the reason that causes this phenomenon. To finalize the above analysis, provide initial possible solutions hopefully leading to useful future works, in Sec. 6 we introduce three kinds of semantic features. All the result numbers reported in the paper support our observations and analysis, and the test result of one of the semantic features concerned by the reviewers is also consistent with our story of helping generalization, and the re-split results are also consistent with our original story.

---

### Author Response · Authors · 2020-05-08
**(FYI) Updated version of this paper has been accepted to IJCAI 2020**

IJCAI 2020 arxiv link: https://arxiv.org/abs/2005.03086

---

### Decision · Program_Chairs · 2019-12-19

**Decision:**

Reject

**Comment:**

The submission is a detailed and extensive examination of overfitting in vision-and-language navigation domains. The authors evaluate several methods across multiple environments, using different splits of the environment data into training, validation-seen, and validation-unseen. The authors also present an approach using semantic features which is shown to have little or no gap between training and validation performance.

The reviewers had mixed reviews and there was substantial discussion about the merits of the paper. However, a significant issue was observed and confirmed with the authors, relating to tuning the semantic features and agent model on the unseen validation data. This is an important flaw, since the other methods were not tuned in this way, and there was no 'test' performance given in the paper. For this reason, the recommendation is to reject the paper. The authors are encouraged to fairly compare all models and resubmit their paper at another venue.